

# Turf performance and physiological responses of native *Poa* species to summer stress in Northeast China

Yajun Chen[1], Zhixin Guo[1], Lili Dong[1], Zhenxuan Fu[1], Qianjiao Zheng[1], Gaoyun Zhang[1], Ligang Qin[2], Xiaoyang Sun[1], Zhenjie Shi[1], Shah Fahad[3], Fuchun Xie[1] and Shah Saud[1]

[1] College of Horticulture and Landscaping, Northeast Agricultural University, Harbin, Heilongjiang, China
[2] College of Animal Science and Technology, Northeast Agricultural University, Harbin, Heilongjiang, China
[3] Department of Agronomy, University of Haripur, Khyber Pakhtunkhwa, Pakistan

## ABSTRACT

Rapid rise in temperature in summer causes severe injury to cool-season turfgrass of both native species and introduced ones in Heilongjiang of Northeast China. The objectives of this study were to compare physiological responses to seasonal heat stresses and turf performances between native and introduced commercial *Poa* accessions. Three Chinese native *Poa* species (*i.e., P. pratensis, P. sibirica* and *P. sphondylodes*) and three USA Kentucky bluegrass cultivars (ie. 'Midnight', 'Moonlight' and 'BlueChip') were evaluated under field conditions in 2017 and 2018. All accessions showed unique characteristics and considerable seasonal differences in response to temperatures. However, performances over all accessions were largely similar in early spring and autumn. In summer, native *P. pratensis* performed similar to 'Midnight', 'Moonlight' or 'BlueChip', with respect to such traits or parameters as quality, coverage, color intensity, growth rate, osmolytes, ROS and anti-oxidant production. Native *P. pratensis* could be used as a new turf resource for further improvement and application under the specific climatic conditions in Heilongjiang; native *P. sphondylodes* may be used in repairing damaged environments or for alternative seasonal greenness.

## INTRODUCTION

A diverse germplasm of the genus *Poa* distributes worldwide in cold and Northern temperate zone, and even high-elevation-tropical regions, where more than 500 species are available in a wide range of habitats (*Soreng, 1990*). Some 105 *Poa* species exist in China, which are mainly distributed in those cold, Northern regions and areas including Qinghai-Tibet plateau (*Chen et al., 2008*). Some species of the genus possess excellent turf attributes, such as fine texture, pleasing color and playability, which are widely exploited and used in sports, public and commercial landscapes, and home lawns (*Chen et al., 2019*). The abundance *Poa* resources in China could potentially be developed for turfgrass and/or for improving resistance to environmental stress. However, there has been no exploitation of theses *Poa* species in Heilongjiang of China yet.

Corresponding authors
Fuchun Xie, xfc204309@163.com
Shah Saud, saudhort@gmail.com

The first Kentucky bluegrass cultivar 'Merion' was released in 1947 by the United States Golf Association (*Meyer, 1982*). Since then, *Poa* breeding activities has been launched in the USA and some European countries, where extensive germplasm collection, subsequent screening and identification morphology, stress physiology and molecular basis, and promoting of elite materials were undertaken (*Saud et al., 2014*; *Saud et al., 2020*; *Johnson et al., 2002*; *Fan et al., 2020*). To date, commercial turf-type species in the genus of *Poa* mainly involve Kentucky bluegrass (*P. pratensis* L.), Canada bluegrass (*P. compressa* L.), perennial biotype of annual bluegrass (*P. annua* spp. reptans), and/or rough bluegrass (*P. trivialis* L.), with Kentucky bluegrass being the most widely used one. Kentucky bluegrass entries are classified into several ecotypes, on the basis of morphology, genetics and resistance to stresses; they include 'Compact', 'Bellevue', 'Mid-Atlantic', 'BVMG', 'Shamrock' and 'America' (*Honig et al., 2012*; *Honig et al., 2018*). Some cultivars were introduced to China in the 1980s; for example, 'Midnight' has been the most widely used one in Northeast China.

Kentucky bluegrass belongs to a monophyletic genus according to geographic evidences (*Hartley, 1961*). It has a high and variable polyploidy, its chromosome numbers range from 5× to 15×, with a base number of seven (*Casler & Duncan, 2003*; *Bushman, Joshi & Johnson, 2018*). However, the number of chromosomes is reported to be of limited value in characterizing Kentucky bluegrass cultivars (*Speckmann & Dijk, 1972*), as genome duplication suggests a link with environmental conditions. Plant polyploids often thrive in harsh and disturbed habitats by alteration of tissue structures and morphological characteristics. Stress response is an important factor in the establishment and success of polyploidy (*Van de Peer et al., 2021*), indicating both the flexibility and the potentiality of *Poa* germplasm improvement in response to different environmental stresses.

Species in genus *Poa* are cool-season grasses, requiring optimum temperatures between 15 °C and 24 °C (*Turgeon, 2008*). Air and/or soil temperatures often reach supra-optimal levels during summer period, which limit shoot and root growth of cool-season grasses leading to plant dormancy or even death (*Fry & Huang, 2004*). Although located in Northeast temperate zone, Heilongjiang province of China is often influenced by monsoon and atmospheric circulation, where heat injury events occur frequently during the seasons of turfgrass growing, especially in the months of June to August. During this period, some introduced cultivars suffer serious damages on turf. Exploitation and utilization of native, wild-type *Poa* species available in Heilongjiang of China and development for resistance to summer heat stress would be effective measures to improve turfgrass quality. Although differential heat resistance strategies have been studied extensively in turfgrass species (*Li et al., 2014*; *Zhang et al., 2017*; *Wang et al., 2018*), limited research has been conducted on evaluating heat resistance of native *Poa* species as well as their comparative heat performances.

Studying turf performances and physiological heat tolerance characteristics in native *Poa* can generate valid data and information to help with planning a plant breeding program on turfgrass improvement. It was extremely hot during June and August in 2017 and 2018 in Heilongjiang, which caused severe heat stress on Kentucky bluegrass turf. Although not what was expected, this event of summer heat stress provided an opportunity for

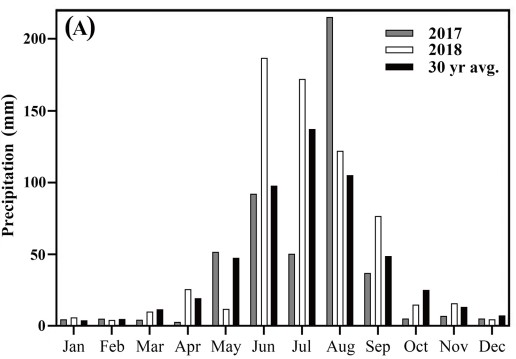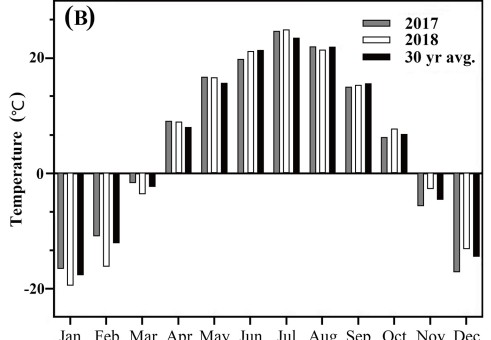

**Figure 1** (A) **Total monthly precipitation (mm) and (B) mean monthly temperature (°C) and 30-yr historical average for Harbin, China, 2017and 2018.** Weather data was obtained from Harbin Meteorological Bureau, Heilongjiang, China.

native *Poa* species to be assessed for turf use. In this research, three native, wild-type *Poa* species (*i.e.*, *P. pratensis, P. sibirica* and *P. sphondylodes*) collected from Heilongjiang province were evaluated and compared with three introduced Kentucky bluegrass cultivars (*i.e.*, 'Midnight', 'Moonlight' and 'BlueChip'). We examined the chromosome numbers for each species and cultivars to elucidate their genetic differences at the cytological level prior to the field experiment. Our hypothesis is that if these selected *Poa* accessions native to Heilongjiang have better summer heat tolerance than introduced cultivars, the native ones could then be used as turfgrass to be adopted locally. The objectives of this study were: (i) to compare seasonal turf performance and physiological heat responses between native, wild-type *Poa* species and commercial Kentucky bluegrass cultivars; and (ii) to determine differential heat resistance characteristics associated with variabilities of these tested accessions.

## MATERIALS AND METHODS

### Experimental site and turf trials

The experiment was conducted over two years (2017–2018) on a black loam soil at the Northeast Agricultural University Horticulture Station in Harbin, Heilongjiang province of Northeast China (45°43′55″N, 126°43′21″E). The research area has a typical cold temperate, continental monsoon climate: hot and rainy summer, and cold and dry winter, with tropical cyclones in summer and autumn seasons. Mean temperatures and precipitation during the study period are listed in Fig. 1. Soil organic matter was 24.6 g/kg with a pH in water of 7.0. Turf was established in May 2016, seeded at 15 g/m$^2$ for complete grow-in to produce a mature turf as the trial was established.

Native species of *P. pratensis*, *P. sibirica* and *P. sphondylodes* (Abbreviated as Hpp, Hpsi and Hpsp, respectively) from Heilongjiang were used as materials, along with three USA commercial Kentucky bluegrass cultivars 'Midnight', 'Moonlight' and 'BlueChip' (Abbreviated as Mid, Moo and Blu, respectively). Natural habitats of 'Hpp', 'Hpsi' and 'Hpsp' are around 45°46′38″N, 126°38′54″E, 45°49′22″N, 127°52′17″E and 46°16′48″N,

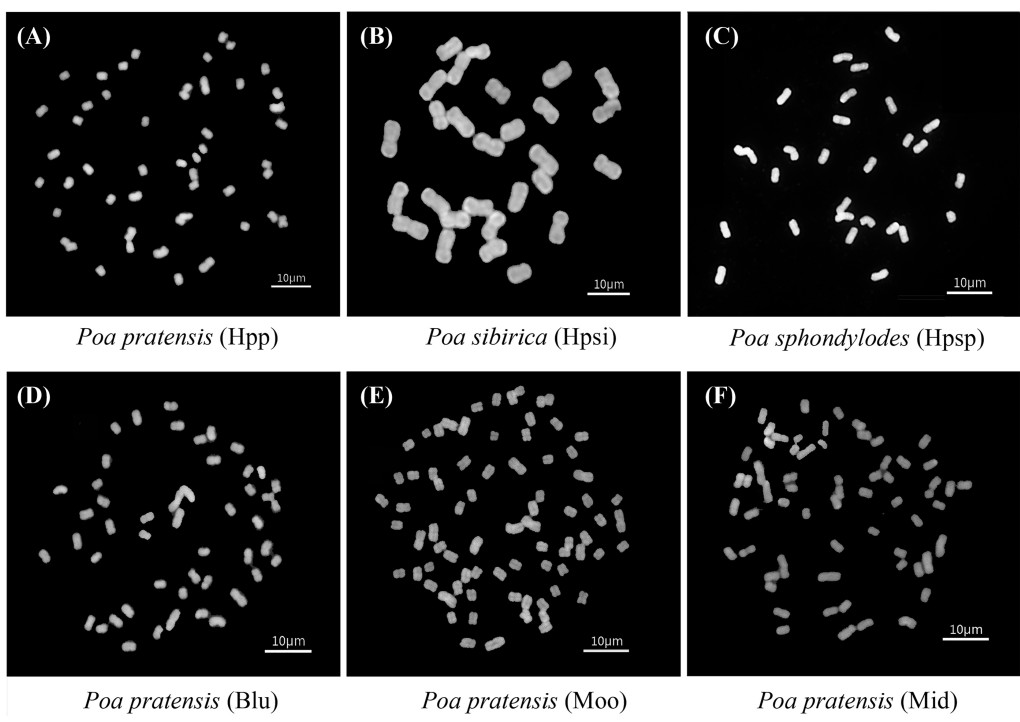

**Figure 2** (A) Metaphase chromosomes of 'Hpp', 2n = 7x = 49; (B) Metaphase chromosomes of 'Hpsi', 2n = 4x = 28; (C) Metaphase chromosomes of 'Hpsp', 2n = 4x = 28; (D) Metaphase chromosomes of 'Blu', 2n = 7x = 49; (E) Metaphase chromosomes of 'Moo', 2n = 11x = 77; (F) Metaphase chromosomes of 'Mid', 2n = 9x = 63. Hpp = *Poa pratensis*; Hpsi = *P. sibirica*; Hpsp = *P. sphondylodes*; Mid = Midnight; Moo = Moonlight; Blu = BlueChip.

124°35′54″E, respectively. Average turf quality (TQ) of 'Mid', 'Moo' and 'Blu' is 6.4, 5.5 and 3.6 reported *earlier* by *Shortell et al. (2004)*. Cytogenetic analyses by FISH indicated differences in chromosomes and ploidy levels of the three native *Poa* species (Fig. 2).

The field trial was conducted using a randomized complete block design with four replicates, with each plot measured 1.2 m wide by 5 m. During the growing seasons, granular and liquid fertilizers were supplied at 2 g N/m² per month. Turfgrass was maintained to a height of 4 cm tall above ground.

## Assessments and measurements

Seasonal measurements were made monthly from May to October in 2017 and 2018. Turf performance was evaluated under natural field conditions, while physiological indices were performed in laboratory on field-collected samples.

## Turf quality, color and turf coverage

Turf quality was rated from 1 to 9, where 1 is very poor, 9 is excellent. Turf color was scaled from 1 to 9, where 1 is completely faded, 9 is the most freshly green. A score of 5 is the lowest acceptable value for both turf quality and color. Turf coverage as a percentage of plot areas was determined monthly.

## Turfgrass growth rate and tiller numbers

The height of turf was recorded on 6th, 7th, 16th, 17th, 26th, 27th of each month throughout the experimental period. The height of grass canopy was measured at 10 sites per plot using a digital caliper (Guanglu, China); growth rate was calculated as the average of the 10 daily height increment (mm) from 4 cm to which it was mowed, two days earlier. Tiller numbers were counted at 6 sites per plot monthly and expressed as numbers of tillers per square centimeters.

## Chromosome

Chromosome numbers in cells of root tips were detected and counted using the fluorescence *in situ* hybridization (FISH) method (*Said et al., 2021*). Briefly, 15 to 20 mm of the end of vigorously growing roots was excised using a scalpel and fixed in a fixative solution of ethanol: glacial acetic acid = 3: 1 (v/v) for 24 h, then transferred to an enzyme mixture (4% cellulase and 2% pectinase), hydrolyzed in a 37 °C water bath, washed with water and fixed, and placed on a glass slide and then dispersed in 45% acetic acid. After drying with a fixative, those well-dispersed slides were selected for FISH using 5S rDNA and 45S rDNA probes. Observations were finally performed with a fluorescence microscope (BX-61, Olympus, Japan), with images collected *via* a microscope camera (DS-Ri1, Nikon, Japan).

## Chlorophyll and osmotic regulating substance

Chlorophyll (Chl) was extracted by soaking 0.1 g of fresh leaves in 10 mL of dimethyl sulfoxide for 72 h (*Hiscox & Israelstam, 1979*); its absorbance (at 646 nm and 663 nm) was recorded with a Micro plate spectrophotometer (Epoch, BioTek, USA).

Content of total soluble protein (TSP) was measured with the method of coomassie brilliant, blue G-250 (*Bradford, 1976*). Powders (2 g in wt) of fresh leaves with 6 ml distilled water were ground into homogenate, and stood in a centrifuge tube for 0.5–1.0 h. Extractions (0.1 ml) were then centrifuged at $4{,}000\times$ g for 15 min at 4 °C, finally, mixed with G-250 (5 ml). Absorbance was read at 646 nm and 663 nm. The content of TSP was calculated according to the standard curve.

Free proline (Pro) content of leaves was determined using the acid ninhydrin method as described by *Ábrahám et al. (2010)* with some modifications. Fresh leaves (0.5 g) were ground in 10 ml of 3% sulphosalicylic acid and centrifuged for 10 min, supernatant (2 g) was added to 3 ml of freshly prepared acid-ninhydrin solution and 2 ml of glacial acetic acid. The mixture was incubated at 90 °C for 1 h and put in an ice bath to stop the reaction; and then, 5 ml of toluene was added to the extract mixture and stood for 20 min at 25 °C to separate toluene and water; finally, the toluene phase was collected to measure the absorbance at 520 nm.

Total soluble sugar (TSS) was measured as described previously (*Wei et al., 2016*) with some modifications. A 0.5 g of powder of fresh leaf was homogenized in 5 ml of 80% ethyl alcohol (v/v); the extraction was centrifuged at $6{,}000\times$ g for 15 min at 4 °C. The separated supernatant was transferred to another test tube, with 12.5 ml of 80% ethyl alcohol (v/v) added. Then, 1 ml of the reaction liquid and 1 ml of 0.2% anthrone were mixed; the

mixture was incubated at 100 °C for 10 min and put in an ice bath for 5 min to stop the reaction. TSS content was determined using a spectrophotometer at 620 nm.

## Antioxidant enzymes activity and $O_2^{\cdot-}$ content

For the assay of antioxidant enzymes activity and level of superoxide anion content ($O_2^{\cdot-}$), 0.5 g of fresh leaves was homogenized at 5–10 °C using 2 mL of 50 mM phosphate extraction buffer (pH 7.8) in an ice-cold mortar. The mixture was centrifuged at 12,000× g for 15 min at 4 °C to collect the supernatant for quantification of enzyme activities and superoxide anion ($O_2^{\cdot-}$). The superoxide dismutase (SOD) activity was measured by recording the rate of p-nitro blue tetrazolium chloride (NBT) reduction in absorbance at 560 nm (*Giannopolites & Ries, 1977*). Superoxide anion content ($O_2^{\cdot-}$) was measured by the hydroxylamine oxidation method and the absorbance was read at 530 nm (*Tian, Gu & Zhu, 2003*). The activity of peroxidases (POD) and catalase (CAT) were determined in absorbance at 240nm and 470 nm, respectively (*Tian, Gu & Zhu, 2003*).

## Levels of glutathione (GSH), ascorbic acid (AsA), malondialdehyde (MDA) and $H_2O_2$

For glutathione (GSH), ascorbic acid (AsA) and malondialdehyde (MDA) contents, 0.1 g of fresh leaves was ground with 1 mL of 10% (w/v) trichloroacetic acid (TCA), centrifuged at 12,000× g at 4 °C for 20 min; then the supernatant was collected for assay. The content of AsA was measured following the method of *Guo et al. (2013)*. 4 ml of $NaH_2PO_4$, 0.4 ml of 10% TCA, 0.4 ml of 44% $H_3PO_4$, 0.4 ml of 4% 2, 2-Bipyridine, and 0.2 ml of 3% $FeCl_3$ were added to 1 ml of supernatant. The mixture was heated at 37 °C for 1 h. Absorbance was measured at 525 nm. Content of GSH was determined according to the method of *Griffith (1980)*, 1 ml of the supernatant was added with 2.4 ml of 0.1 mol $L^{-1}$ phosphate buffered saline (pH 7.7) and 0.2 ml of 4 mmol $L^{-1}$ 5, 5-dithiobium-(2-nitrobenzoic acid). Absorbance was determined at 412 nm. Content of MDA was determined by the thiobarbituric acid method (*Ma et al., 2015*), 2 ml of the supernatant was mixed with 2 ml of 0.6% TBA, the mixture was then heated at 95 °C for 30 min, quickly cooled, and then centrifuged at 10,000× g for 10 min. Absorbance of the supernatant was measured at 450 nm, 532 nm, and 600 nm to calculate the MDA content. The procedure outlined in the $H_2O_2$ Quantitative Assay Kit (BioVision, USA) was followed for the assay of $H_2O_2$ content at 415 nm.

## Statistical analysis

Mean data over two years of 2017 and 2018 in projected turf performance and physiological indices were analyzed using ANOVA (SPSS v 18.0, Inc., Chicago IL). The Fisher's protected least significant difference (LSD) at $P < 0.05$ was used to detect if significant differences existed among entries or among months in the experiment.

**Table 1 The monthly turf quality and color responses of three native Poa species and three introduced cultivars of Kentucky bluegrass during the 2017 and 2018 growing seasons in Harbin, China.**

| Entries | Two years average turfgrass quality (1–9) | | | | | | Two years average turf color (1–9) | | | | | |
|---|---|---|---|---|---|---|---|---|---|---|---|---|
| | May | Jun | Jul | Aug | Sept | Oct | May | Jun | Jul | Aug | Sept | Oct |
| Hpp | 7.5abA | 7.2bB | 6.8abC | 6.5abC | 7.0bBC | 7.8aA | 8.5aA | 8.0aAB | 7.5abB | 6.8aC | 7.5bB | 8.2abA |
| Hpsi | 6.4cA | 6.0dA | 5.4cB | 5.0cB | 5.5cB | 6.2cA | 6.5cA | 6.2cA | 5.5cB | 5.5bB | 6.6dA | 6.8cA |
| Hpsp | 7.8abA | 4.0eC | 1.0dC | 1.0dC | 4.5dC | 7.0bB | 8.5aA | 5.0dB | 1.0dC | 1.0cC | 8.8aA | 8.6aA |
| Mid | 8.0aA | 7.8aAB | 7.5aB | 7.0aC | 7.8aAB | 8.0aA | 8.5aA | 8.2aA | 8.0aA | 7.0aB | 7.8bA | 8.5aA |
| Moo | 7.2bA | 7.0bcAB | 6.5bBC | 6.0bC | 7.2bA | 7.5aA | 8.0bA | 7.8aAB | 7.5abAB | 6.4abC | 7.4bcB | 7.8bAB |
| Blu | 6.8bcAB | 6.5cB | 6.0bC | 5.2cD | 6.0cC | 7.0bA | 8.0bA | 7.2bB | 7.0bB | 5.8bC | 7.0cdB | 7.5bcAB |

Notes.
Hpp, *Poa pratensis*; Hpsi, *P. sibirica*; Hpsp, *P. sphondylodes*; Mid, Midnight; Moo, Moonlight; Blu, BlueChip.
Lowercase letters denote significant differences within each vertical row for dif-ferent entries ($P < 0.05$); upper case letters indicate significant differences within each horizontal row for different months in one entry ($P < 0.05$).

## RESULTS

### Turfgrass quality and color

Results of effects of seasonal temperature on turf quality (TQ) and color of the native and introduced bluegrasses are presented in Table 1. TQ of 'Mid' was statistically superior throughout the seasons but similar to that of native 'Hpp' except during September.

TQ of 'Moo' was similar to that of 'Mid' during September and October. 'Blue' performed similar to native 'Hpsi' throughout the growing seasons. During May, September and October, the turf color of 'Hpsp' being darker green was significantly better to that of 'Moo', 'Blu' or 'Hpsi', but similar to that of 'Mid', 'Moo' and 'Hpp'. Whilst TQ was the poorest, compared with other bluegrasses from June to September. During early spring seasons, the color of 'Hpsp' was significantly better than that of 'Hpsi' and 'Blu', but similar to that of 'Moo' and 'Blu' in October. In summer, the color of 'Hpp' (8.0 in June, 7.5 in July, and 6.8 in August) was similar to that of 'Mid' and 'Moo'.

### Turf coverage and growth rate

Analysis of variance exhibited significant variabilities in turf coverage and growth rate among accessions varying with changing temperatures (Table 2). Maximum turf coverage was observed in 'Mid', it was however statistically similar-to that in 'Moo' and 'Hpp'. Turf coverage observed in 'Blu' was initially similar–to that in 'Moo', 'Mid' and 'Hpp' in the month of May but was lower than in 'Mid' in other subsequent months of the growing season. Although the turf coverage of 'Hpsi' performed lower than the rest except 'Hpsp', that was still acceptable. The lowest turf coverage was observed in 'Hpsp' from June to August. Generally, the turf coverage reduced in summer and then increased in autumn. The highest growth rate was observed in 'Mid' regardless of a rising seasonal temperature from early spring to summer, which was similar to that in native 'Hpsi' except September. The growth rate measured in 'Hpp' was similar to that in 'Moo' through the growing season, and significantly lower than that in 'Mid' and 'Hpsi' but generally higher than that in 'Blu'. 'Hpsp' had the lowest growth rate during spring and autumn.

**Table 2** The monthly turf coverage and growth rate responses of three native Poa species and three introduced cultivars of Kentucky bluegrass during the 2017 and 2018 growing seasons in Harbin, China.

| Entries | Two years average turfgrass coverage (%) | | | | | | Two years average turfgrass growth rate (mm/per day) | | | | | |
|---------|------|------|------|------|------|------|------|------|------|------|------|------|
|  | **May** | **Jun** | **Jul** | **Aug** | **Sept** | **Oct** | **May** | **Jun** | **Jul** | **Aug** | **Sept** | **Oct** |
| Hpp | 94.4aAB | 96.4abA | 94.1abAB | 86.4bC | 90.5bBC | 93.2aAB | 11.4bB | 12.8bA | 10.4bB | 8.6bC | 10.2bB | 10.8bcB |
| Hpsi | 85.5cB | 88.5cA | 81.6cD | 83.2cC | 85.2bB | 84.7cB | 17.8aAB | 18.6aA | 16.3aB | 12.5aD | 13.8aCD | 14.6aC |
| Hpsp | 80.6dB | 30.6dC | 5.5dD | 3.2dD | 77.6cB | 88.5bcA | 4.8dA | 3.4dB | — | — | 2.6dC | 4.6dA |
| Mid | 96.8aA | 98.5aA | 96.2aA | 90.3aB | 95.8aA | 95.5aA | 17.3aAB | 18.9aA | 16.3aB | 12.7aD | 14.8aB | 15.3aB |
| Moo | 93.5abA | 94.2bA | 91.7abA | 86.5bC | 89.5bB | 93.6aA | 11.2bcB | 12.5bA | 9.6bcC | 8.1bD | 10.7bBC | 11.2bB |
| Blu | 90.3bA | 90.8bcA | 88.4bcB | 84.3cC | 88.2bB | 90.4abA | 9.9cAB | 10.6cA | 8.4cBC | 7.8bC | 9.1cB | 9.6cA |

**Notes.**

Hpp, *Poa pratensis*; Hpsi, *P. sibirica*; Hpsp, *P. sphondylodes*; Mid, Midnight; Moo, Moonlight; Blu, BlueChip.
Lowercase letters denote significant differences within each vertical row for dif-ferent entries ($P < 0.05$); upper case letters indicate significant differences within each horizontal row for different months in one entry ($P < 0.05$).

**Table 3** The tiller number responses of three native Poa species and three introduced cultivars of Kentucky bluegrass during 2017 and 2018 growing seasons in Harbin, China.

| Entries | Two year average turfgrass tillers (tillers/cm$^2$) | | | | | |
|---------|------|------|------|------|------|------|
|  | **May** | **Jun** | **Jul** | **Aug** | **Sept** | **Oct** |
| Hpp | 10.8bC | 11.9bB | 12.8abAB | 11.9aB | 12.4bAB | 12.9bA |
| Hpsi | 7.3dC | 9.0cA | 9.5cA | 6.9cC | 8.3dB | 8.9cAB |
| Hpsp | 5.4eAB | 6.1dA | 1.7dD | 1.3dD | 2.7eC | 5.1dB |
| Mid | 11.9aD | 12.9aBC | 13.2aAB | 12.5aCD | 13.4aAB | 14.0aA |
| Moo | 10.2bcB | 12.1abA | 12.6abA | 10.8bB | 12.1bA | 12.5bA |
| Blu | 9.7cC | 11.4bAB | 12.2bA | 10.4bC | 11.3cB | 12.2bA |

**Notes.**

Hpp, *Poa pratensis*; Hpsi, *P. sibirica*; Hpsp, *P. sphondylodes*; Mid, Midnight; Moo, Moonlight; Blu, BlueChip.
Lowercase letters denote significant differences within each vertical row for dif-ferent entries ($P < 0.05$); uppercase letters indi-cate significant differences within each horizontal row for different months in one entry ($P < 0.05$).

## Number of tillers

The number of tillers were significantly different among entries in both years (Table 3). Maximum tillers were observed in 'Mid' regardless of changes of the seasonal temperatures. The tiller number of 'Hpp' was statistically similar to that of 'Mid' and 'Moo' for the whole growing period although slightly different in May and September. A similar seasonal trend in tiller numbers was observed in 'Hpsi' during both years, however, its tiller numbers were apparently lower than those of others except 'Hpsp' which had the lowest tiller numbers among all entries.

## H$_2$O$_2$, O$_2^{\cdot-}$, Chl and MDA contents

H$_2$O$_2$, O$_2^{\cdot-}$, Chl and MDA contents among the turfgrass entries showed significant differences (Figs. 3A–3D). These parameters were observed to increase with rising temperatures from spring to summer, reaching a peak in August and then declining from October. Overall, H$_2$O$_2$ and O$_2^{\cdot-}$ contents of native accessions were significantly higher than those of introduced cultivars in spring and early summer. A maximum production of H$_2$O$_2$ and O$_2^{\cdot-}$ was observed in 'Hpsp' in May and June, which was surpassed by 'Hpsi'

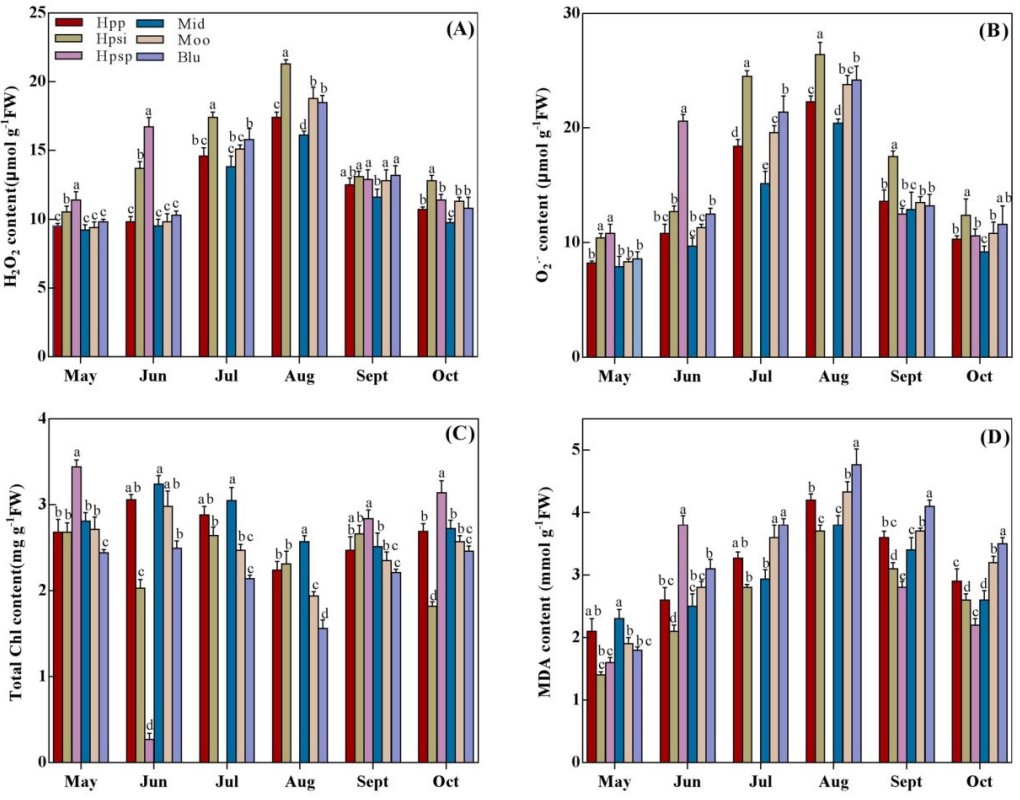

**Figure 3** (A–D) The $H_2O_2$, $O_2^{\cdot-}$, total chlorophyll and MDA of six *Poa* accessions during the growing seasons of 2017 to 2018. Error bars represent 95% confidence intervals and lowercase letters denote significant differences ($P < 0.05$); Hpp = *Poa pratensis*; Hpsi = *P. sibirica*; Hpsp = *P. sphondylodes*; Mid = Midnight; Moo = Moonlight; Blu = BlueChip.

in summer and autumn. The lowest $H_2O_2$ and $O_2^{\cdot-}$ contents were observed in 'Mid'. However, it was statistically similar to those in 'Hpp', 'Moo' or 'Blu', from May to July but notably lower in the rest of the experimental period (Jul.–Oct.). MDA content varied significantly among species with changing temperatures (Fig. 3D). The MDA concentration in all tested accessions was lower in spring, increasing in summer while declining in autumn. Among the entries, the maximum MDA was observed in 'Mid' in early spring, which was similar to 'Hpp'. In summer and late autumn, the highest MDA concentration was noted for 'Blu' which was similar to that for 'Moo' and 'Hpp' while the lowest was observed in 'Hpsi'. The Chl content in all tested accessions was also significantly affected by seasonal temperatures (Fig. 3C). The highest Chl content was observed in 'Hpsp' in early spring, which decreased in summer and recovered in autumn. Chl content in 'Mid' performed similar to that in 'Hpp'. Chl content in 'Blu' was distinctly lower in early spring than that in other entries while was similar to that in 'Moo' in summer and autumn. It was also noted that Chl content in 'Moo' and 'Hpp' was similar throughout the experiment.

## SOD, CAT, POD, and GSH activity

SOD, CAT, POD, and GSH activities are presented in Fig. 4. The SOD activity showed an increasing trend with rising temperature from spring to summer and then declining from autumn (Fig. 4A). The highest SOD activity was observed in 'Mid' while the lowest was recorded in native 'Hpsp' regardless of the seasonal temperatures. The SOD activity in 'Hpp' was statistically similar to that in 'Mid' and 'Moo' in late summer and autumn-similar to that in 'Moo' and 'Blu' in early summer while higher than that in 'Hpsi' and 'Hpsp' for the experimental duration. The CAT activity varied significantly among the accessions, but, with no effect of seasonal temperatures (Fig. 4B). The highest CAT activity was observed in 'Mid' which was similar to that in 'Hpsi', and so were in 'Moo' and 'Hpp' across the seasons. The lowest CAT activity was observed in 'Hpsp' in spring and autumn. The POD activity was lower in spring while it increased with temperatures in summer (Fig. 4C). The Maximum POD ativity was observed in 'Hpp' in all seasons. The POD activity observed for 'Mid' was comparable to 'Hpp' for the experimental duration except in May, when it was slightly lower. 'Hpsp' showed the lowest POD activity followed by 'Hpsi'. Similarly, 'Hpp' showed a GSH activity comparable to 'Mid' in early summer and autumn, which was clearly higher than 'Hpsi' and 'Hpsp' but similar to 'Moo' and 'Blu'. A lower GSH activity was detected in 'Hpsp' regardless of seasonal temperatures (Fig. 4D).

## TSP, TSS, Proline and AsA content

Significant variations ($P < 0.05$) in TSP, TSS, Proline and AsA content among *Poa* accessions under changing seasonal temperatures (Fig. 5). 'Hpp' had a TSP content similar to 'Mid' for most of the experimental period except late summer, when 'Hpp' and 'Hpsi' were higher than 'Mid' (Fig. 5A). 'Moo' performed similar to 'Mid' and 'Hpp' in spring and early summer, but lower in late summer and autumn, which were however similar to 'Hpsp'. and 'Blu'. The TSS content also increased with rising temperatures from spring to summer and then declined in autumn (Fig. 5B). 'Mid' produced the maximum TSS, which was similar to 'Hpp' throughout the experiment except late summer when 'Hpp' was distinctly greater than 'Mid'. 'Blu' was comparable to 'Moo' and 'Hpp' across all the seasons. The lowest TSS production was observed in 'Hpsi'. The accessions varied significantly in proline content with changing temperatures from spring to autumn (Fig. 5C). Proline concentration increased in 'Mid', 'Moo', and 'Hpp' more than other individual *Poa* entries. The performance among entries was mostly indistinguishable in spring and early summers but onward, 'Hpsi' and 'Hpsp' lost its ability to produce proline and showed the lowest proline concentration. The AsA content differed significantly among the accessions (Fig. 5D). An increase in AsA content was more prominent in 'Mid' followed by 'Hpp' from spring to summer. In spring, the AsA produced by 'Hpp', 'Mid', 'Moo' or 'Blu' was roughly the same, but from July to October, maximum AsA was detected in 'Mid' followed by 'Hpp', 'Moo' and 'Blu'.The lowest AsA concentration was observed in 'Hpsp'.

## DISCUSSION

Heat stress can adversely affect the growth of cool-season turfgrasses and then reduce the turf quality in Heilongjiang, China. We observed significant variabilities among *Poa*

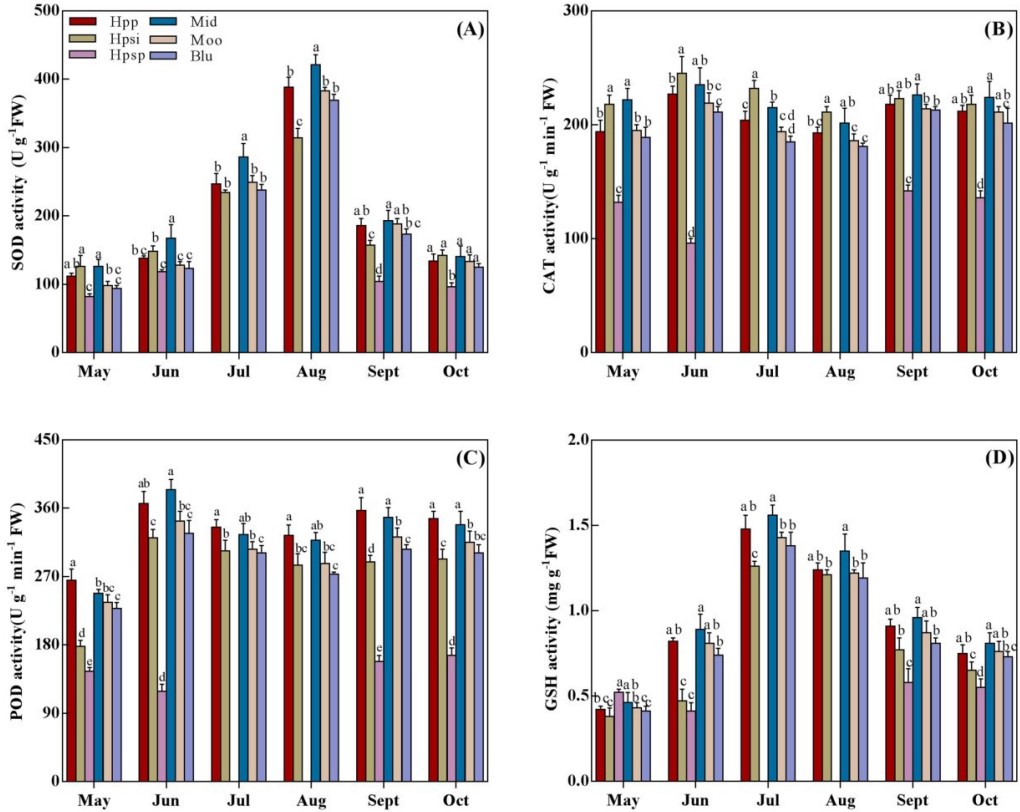

**Figure 4** (A-D) **The SOD, CAT, POD, and GSH activity of six *Poa* accessions during the growing seasons of 2017 to 2018.** Error bars represent 95% confidence intervals and lowercase letters denote significant differences ($P < 0.05$); Hpp = *Poa pratensis*; Hpsi = *P. sibirica*; Hpsp = *P. sphondylodes*; Mid = Midnight; Moo = Moonlight; Blu = BlueChip.

accessions for growth, TQ, and turf color during the summer season. For both native and introduced accessions, TQ, color, coverage and growth rate began to drop with rising temperatures in summer, in comparison with those in early spring and late autumn (Tables 1 and 2), but the variations of thermo-tolerance varied with accessions, indicating that heat resistance and adaptation of these grasses were species-specific. Compared with introduced cultivars, native 'Hpp' performed well, was equal to 'Mid' and 'Moo' for resistance to heat stress in both summers of year 2017 and 2018; and was more heat-tolerant than 'Blu'. 'Hpsi' and 'Hpsp' responded differently, subject to certain physiological and morphological parameters and seasons. These accessions differ in chromosome numbers and ploidy (Fig. 2).

'Hpsp' and 'Hpsi' were detected to be tetraploids ($2n = 4x = 28$); collections of Kentucky bluegrass were anisopolyploids with chromosome numbers of 49 (or 7x), 63 (or 9x), 77 (or 11x) and 49 (or 7x), respectively, corresponding to 'Hpp', 'Mid', 'Moo' and 'Blu', hence these accessions are categorized to specific biological species in *Poa* genus. Correlations between heat stress and chromosome numbers in *Poa* are unknown, although a previous study mentioned that polyploidy confers a selective advantage under stressful or

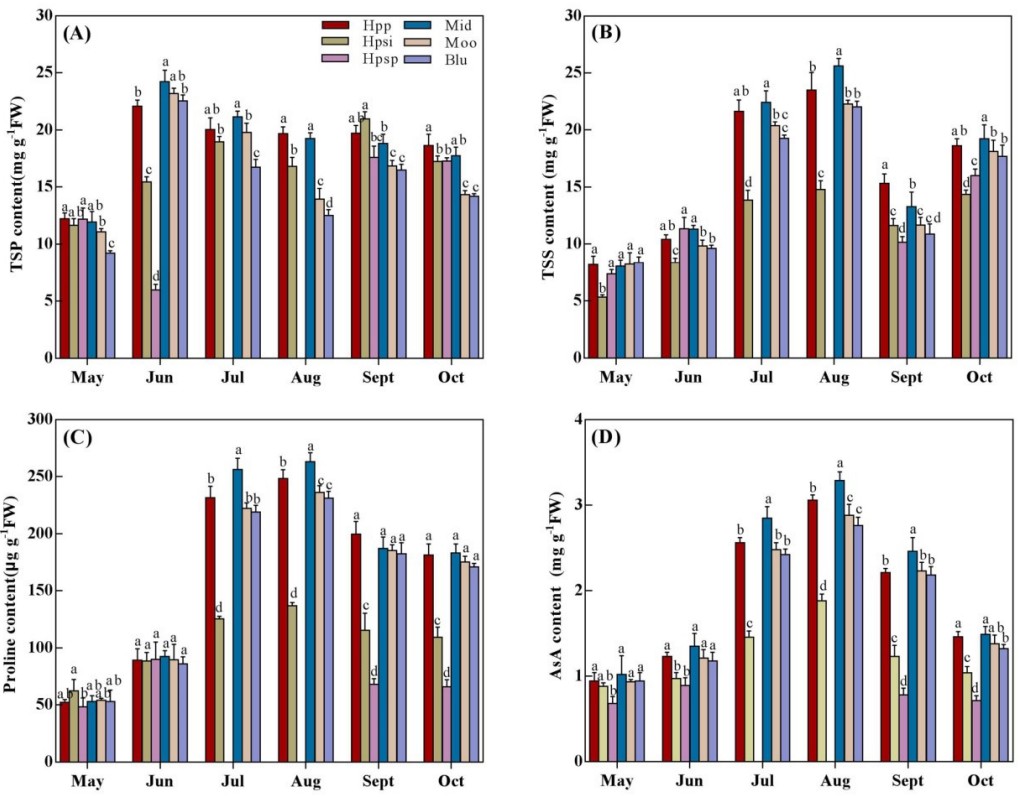

**Figure 5** (A-D) The TSP, TSS, Proline and ASA content of six *Poa* accessions during the growing seasons of 2017 to 2018. Error bars represent 95% confidence intervals and lowercase letters denote significant differences ($P < 0.05$); Hpp = *Poa pratensis*; Hpsi = *P. sibirica*; Hpsp = *P. sphondylodes*; Mid = Midnight; Moo = Moonlight; Blu = BlueChip.

changing environmental conditions (*Van de Peer, Mizrachi & Marchal, 2017*). Using QTL and bio-omics techniques to reveal the relationship between ploidy and heat resistance of *Poa* could be of a great significance (*Estelle et al., 2021*). Interestingly, species 'Hpsp' had characters showing that mature seeds shed naturally from panicles onto ground in June; and that these seeds remained dormant in summer, which led to no growth, fading on turf color, and nearly no turf coverage during July and August. However, after summer, 'Hpsp' recovered from dormancy, seeds started germination and tillers developed, with unique dark green color in September and October. These attributes of 'Hpsp' should be useful in repairing damaged environments or for alternative seasonal greenness with lower input. Compared with 'Hpsp', 'Hpsi' did not possess these characteristics in summer, but displayed a poorer turf quality than others.

Avoiding leaf senescence is critical in maintaining visual turf quality under heat stress. Chlorophyll is an important leaf senescence indicator of heat stress, which reduces the turf quality. A previous study indicated that heat stress leads to reductions in the content of Chl in Kentucky bluegrass that could be associated with the production of reactive oxygen species (ROS) in response to heat stress (*He & Huang, 2010*). Heat stress induces ROS

including $H_2O_2$, $O_2^{\cdot-}$, $HO^-$, $O_2^1$, which are partially reduced forms of atmospheric oxygen ($O_2^{\cdot-}$), and lead to oxidative destruction of cell and pigment breakdown (*Mitter, 2002*; *Abraham et al., 2004*; *Bita & Gerats, 2013*; *Yang et al., 2014*). To minimize the oxidative damage under heat stress, plants use antioxidants to dispose of ROS, including enzymatic and nonenzymatic constituents (*Bowler, Montagu & Inze, 1992*; *Xu et al., 2006*; *Du, Wang & Huang, 2009*). $H_2O_2$, $O_2^{\cdot-}$ and MDA as ROS representatives and some ROS-Scavenging enzymes including SOD, POD, CAT and GSH were investigated in 2017 and 2018 for evaluating heat-resistant ability and potential of selected accessions. We observed that $H_2O_2$, $O_2^-$ and MDA increased in all accessions with rising temperatures in summer (Figs. 3A, 3B, 3D). However, Kentucky bluegrass 'Hpp', 'Mid', 'Moo' and 'Blu' exhibited a lower content of $H_2O_2$, $O_2^-$ and MDA than 'Hpsi' and 'Hpsp' in June, or than 'Hpsi' in July and August. On the contrary, ROS scavenging enzymes activities of SOD, POD, CAT and GSH were more vigorous in 'Hpp', 'Mid', 'Moo' and 'Blu' than in 'Hpsi' and 'Hpsp', suggesting that the antioxidant activity of 'Hpp', 'Mid', 'Moo' and 'Blu' was comparatively more than that of 'Hpsi' and 'Hpsp', due to enhanced production of antioxidant (Figs. 4A–4D). These results are in agreement with *Smirnoff (1993)* who reported that, plants use ROS scavenging mechanisms to protect cells from oxidative injury, including the activation of antioxidant enzymes to break down $H_2O_2$ to water. We also found that the capacity of oxidative scavenging and accumulation cellular osmolytes such as TSP and proline differed among Kentucky bluegrass accessions, indicating that some physiological responses to heat varied within cultivars. According to *Dat et al. (2000)*, oxidative scavenging capacity differs between plant species and cultivars under varying environmental stress conditions. A positive association between antioxidant activities and heat tolerance have also been documented in other cool-season turfgrass or cultivars in Kentucky bluegrass (*Scandalio, 1993*; *Scandalio, 1993*; *Zhang & Schmidt, 2000*; *Jiang & Huang, 2001*). However, in most cases, native, wild-type 'Hpp' and introduced 'Mid' showed similar performance in heat resistance in terms of plant growth, tiller numbers and physiological response, suggesting that 'Hpp' has a great potential to be developed into an excellent variety for commercial use in Heilongjiang, China.

## CONCLUSION

Summer heat stress occurs in Heilongjiang of Northeast China even though it is located in the temperate zone; the stress affects cool-season turfgrass growth and turf quality. Exploring native resources of heat-resistant turfgrass is important to urban landscaping nowadays and in the future.

Heat stress-related physiological parameters and plant growth performance were studied on three native *Poa* species 'Hpp', 'Hpsp' and 'Hpsi', in comparison with three introduced Kentucky bluegrass cultivars 'Mid', 'Moo' and 'Blu'. As expected, native 'Hpp' performed mostly comparable to introduced accessions; showed similar characteristics in heat resistance in turf quality, growth rate, tiller numbers and physiological responses such as ROS, osmolytes and anti-oxidant production, suggesting that native 'Hpp' could be used as a new turf resource for further improvement and application under the specific climatic

condition of Heilongjiang. 'Hpsp' may be used in repairing damaged environments or for alternative seasonal greenness, based on its uniquely dark green color in spring and autumn and its recovery ability from summer. However, 'Hpsi' is not recommended as turf resource for use.

## ACKNOWLEDGEMENTS

We thank the editor and reviewers for helpful comments. We thank Dr. Xianguang Zhang from Advanta seeds Thailand for his great help to improve the manuscript.

### Funding

This research was funded by the National Natural Science Foundation of China (No 31971772; 32001407) and the College Student Innovation and Entrepreneurship Training Program at Northeast Agriculture University. The funders had no role in study design, data collection and analysis, decision to publish, or preparation of the manuscript.

### Grant Disclosures

The following grant information was disclosed by the authors:
The National Natural Science Foundation of China: 31971772, 32001407.
The College Student Innovation and Entrepreneurship Training Program at Northeast Agriculture University.

### Competing Interests

The authors declare there are no competing interests.

### Author Contributions

- Yajun Chen, Xiaoyang Sun, Zhenjie Shi and Shah Saud conceived and designed the experiments, prepared figures and/or tables, and approved the final draft.
- Zhixin Guo and Zhenxuan Fu conceived and designed the experiments, performed the experiments, prepared figures and/or tables, and approved the final draft.
- Lili Dong conceived and designed the experiments, performed the experiments, authored or reviewed drafts of the paper, and approved the final draft.
- Qianjiao Zheng, Gaoyun Zhang and Ligang Qin conceived and designed the experiments, analyzed the data, prepared figures and/or tables, and approved the final draft.
- Shah Fahad and Fuchun Xie conceived and designed the experiments, prepared figures and/or tables, authored or reviewed drafts of the paper, and approved the final draft.

### Data Availability

Raw data are available as a Supplemental File.

### Supplemental Information

Supplemental information for this article can be found online at http://dx.doi.org/10.7717/peerj.12252#supplemental-information.

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
