# Peer review of "Turf performance and physiological responses of native Poa species to summer stress in Northeast China"

_PeerJ, doi:10.7717/peerj.12252_

## Round 0.1 · original submission · Major Revisions

Dear Dr. Chen,

Thank you for your submission to PeerJ.

It is my opinion as the Academic Editor for your article - Turf performance and physiological responses to summer stress for native Poa species in Northern China - that it requires a number of Major Revisions.

My suggested changes and reviewer comments are shown below and on your article 'Overview' screen.

Please address these changes and resubmit.

Reviewer 1 ·

Basic reporting

The methodic should be rewrite to be more clearer both to the reviewer and the reader.

Scientific background of the topic is well-provided.

Experimental design

In abstract Authors should change time of evaluated years - all experiments refers only to 2017 and 2018, not 2016.

The cytogenetic analysis by FISH before this study indicated the differences in chromosomes and ploidy levels of three native Poa accessions (Fig. 2) - Please explain are these photos used originally or already published in other scientific paper? Manuscript lacks of methods used to obtain photos of chromosome plates. Are these foreign cultivars also polyploids? Manuscript lacks of chromosome plates or cytogenetic information about used 3 commercial cultivars.

Titles of Fig. and 5 should be edited, there are some mistakes or they seem to be uncompleted.
Title of Fig. 4 if Authors are using Hpp abbreviation for Poa pratensis then all species should abbreviated in Fig. 3 to 5.

Validity of the findings

Impact and novelty in the manuscript is assessed.

Not all underlying data have been provided and should be completed.

Introduction, discussion and conclusions are well stated but should be rewritten. I suggest to expand information about how ploidy levels determine plant size and its potential response to environmental stress. All these information should be addressed to introduction and discussion part after completing the manuscript with cytogenetic information about foreign cultivars.

Additional comments

The manuscript should be completed of cytogenetic data and expanded of ploidy levels information in all proper sections.

Reviewer 2 ·

Basic reporting

The manuscript entitled "Turf Performance and Physiological Responses to Summer Stress for NativePoa Species in Northern China" is an exprtmental work in the field of stress physiology, and specifically in this case temperature stress is taken into account.
(1) However, since the authors conducted a microfield experiment in accordance with the assumed experimental system, I wonder on what basis they refer to a single stress factor in the manuscript presented for evaluation.
(2) The text contains a few linguistic disabilities, and as an example I will cite the second sentence of the Abstract, which probably does not start well and perhaps should better be: The objective of the study were to compare ....
(3) There is no formulation of a research hypothesis by the authors in the Introduction
(4) Not all literature citations in the text are reflected in References - lines 269, 300, 331
Moreover, the form of citation in the text is not standardized, e.g. lines 48 and 51 and so on
(5) Overall the structure of the manuscript is correct , but its content should be corrected.

Experimental design

Research questions are not clearly presented, even if the problem raised in the paper is significant
The experimental setup should be presented a bit more precisely. All the physiological methods used must be briefly described. It is not enough to provide a reference to the literature - the reader cannot be forced to look for information in the literature resources of other journals .

Validity of the findings

The results obtained by the authors are interesting and original. Their significance is local, but they may also be of interest to a wider audience. The problem lies in the difficulty of tracking the data contained in Figures 3, 4, 5. Can you perhaps use colored bars?
The discussion is definitely room for improvement. Discusiion is too long and not very specific.
From the References section, I would remove publications that have a rather loose connection with the topic of the work.

Additional comments

The work certainly has potential, but it should be refined, improved in the sense of clarifying the message. It will be possible if the research hypotheses are initially formulated and in the methodological part the authors specify the plan of the experimental system. I suggest a diagram. Then please shorten the discussion of the obtained results to the most important threads. The quality of the text and its message are important, not the number of pages saved.

---

## Round 0.2 · Minor Revisions

Dear Authors,
Please, correct the manuscript according to Reviewer 3's suggestions. Please, correct also the language.

Reviewer 1 ·

Basic reporting

The article is well rewritten. It is now much easier to follow every part.

The discussion is much better, data-based and consistent with the findings.

Experimental design

The figures are now much more informative and the methods are properly rewritten.

Validity of the findings

The authors have satisfactorily responded to all my and especially to the other Reviewer's questions and made the necessary changes to the manuscript.

Reviewer 3 ·

Basic reporting

The authors wrote Turf Performance and Physiological Responses to Summer Stress for Native Poa Species in Northern China, in my opinion, this represents a very interesting topic, and the manuscript however shows some weaknesses which require a further review and a careful revision of the language through the text.

Experimental design

The design used is well suitable.

Validity of the findings

The results are vailed and may be published.

Additional comments

1) The cytogenetic analysis by FISH before this study indicated the differences in chromosomes and ploidy levels of three native Poa accessions (Fig. 2) - Please explain are these photos used originally or already published in other scientific paper?

2) There is no formulation of a research hypothesis by the authors in the Introduction.

3) Manuscript lacks of methods used to obtain photos of chromosome plates. Are these foreign cultivars also polyploids? Manuscript lacks of chromosome plates or cytogenetic information about used 3 commercial cultivars.
4) Research questions are not clearly presented, even if the problem raised in the paper is significant.

5) Not all literature citations in the text are reflected in References - lines 269, 300, 331.

6) From the References section, I would remove publications that have a rather loose connection with the topic of the work.

7) References
Standardize references

---

## Round 0.3 · accepted · Accept

Dear Dr. Chen,
Thank you for your submission to PeerJ.
I am writing to inform you that your manuscript - Turf Performance and Physiological Responses of Native Poa Species to Summer Stress in Northeast China - has been Accepted for publication. Congratulations!
This is an editorial acceptance; publication is dependent on authors meeting all journal policies and guidelines.


Gearard also noted the following minor edits that you should make during the proofing process:

EDITS
LINE NO: / BEFORE / AFTER / [COMMENTS]
LINE 39: / molecular bases / molecular basis / [.]
LINE 45: / resistance to stresses, they / resistance to stresses; they / [.]
LINE 47: / 1980s, for example, ‘Midnight’, which has / 1980s; for example, ‘Midnight’ has / [.]
LINE 75: / not what we expected / not what was expected / [.]
LINE 76: / species to be assesses for / species to be assessed for / [.]
LINE 90: / (20172018) / (2017-2018) / [.]
LINE 209: / similarto / similar to / [.]
LINE 220: / years Table 3). / years (Table 3). / [.]
LINE 223: / trend in tillers / trend in tiller / [.]
LINE 225: / lowest tillers numbers / lowest tiller numbers / [.]
LINE 251: / autumn;similar / autumn; similar / [.]
LINE 255: / in ‘Hpsi’, so were in ‘Moo’ / in ‘Hpsi’, and so were in ‘Moo’ / [.]
LINE 257: / while increased with / while it increased with / [.]
LINE 259: / it was slightly low. / it was slightly lower. / [.]
LINE 275: / ‘Mid’, ‘Moo’, or ‘Hpp’ more / ‘Mid’, ‘Moo’, and ‘Hpp’ more / [.]
LINE 294: / ploidies / ploidy / [.]
LINE 358: / ACKNOMLEDGEMENTS / ACKNOWLEDGEMENTS / [.]